# An Overview of Head and Neck Tumor Reirradiation: What Has Been Achieved So Far?

**DOI:** 10.3390/cancers15174409

**Published:** 2023-09-04

**Authors:** Konstantin Gordon, Daniil Smyk, Igor Gulidov, Kirill Golubev, Timur Fatkhudinov

**Affiliations:** 1A. Tsyb Medical Radiological Research Center, Branch of the National Medical Research Radiological Center of the Ministry of Health of the Russian Federation (A. Tsyb MRRC), 4, Korolev Street, 249036 Obninsk, Russia; smyk-di@rudn.ru (D.S.); iagulidov@mrrc.obninsk.ru (I.G.); doc.golubevkirill@yandex.ru (K.G.); 2Medical Institute, Peoples’ Friendship University of Russia (RUDN University), Miklukho-Maklaya Street 8, 117198 Moscow, Russia; fatkhudinov-tkh@rudn.ru

**Keywords:** reirradiation, head and neck cancer, IMRT, VMAT, SBRT, brachytherapy, proton therapy, carbon ions, neutrons

## Abstract

**Simple Summary:**

Head and neck cancers are highly prevalent worldwide. Despite the recent advances in cancer diagnosis and treatment, more than 50% of the cases relapse within 2 years after the initial treatment. Radiation therapy, a leading option in the head and neck tumor management, has increasing potential as a repeated treatment. This review outlined recent advances in radiotherapy as applied to recurrent head and neck cancers in terms of clinical outcomes.

**Abstract:**

The recurrence rate of head and neck cancers (HNCs) after initial treatment may reach 70%, and poor prognosis is reported in most cases. Curative options for recurrent HNCs mainly depend on the treatment history and the recurrent tumor localization. Reirradiation for HNCs is effective and has been included in most guidelines. However, the option remains clinically challenging due to high incidence of severe toxicity, especially in cases of quick infield recurrence. Recent technical advances in radiation therapy (RT) provide the means for upgrade in reirradiation protocols. While the majority of hospitals stay focused on conventional and widely accessible modulated RTs, the particle therapy options emerge as tolerable and providing further treatment opportunities for recurrent HNCs. Still, the progress is impeded by high heterogeneity of the data and the lack of large-scale prospective studies. This review aimed to summarize the outcomes of reirradiation for HNCs in the clinical perspective.

## 1. Introduction

Head and neck cancers (HNCs) are rated 6–7 in global prevalence among malignant tumors, with 700,000 newly registered cases and 470,000 deaths a year [1,2]. The disease group includes malignant tumors of the tongue, oral cavity, lips, nasal cavity, nasopharynx, oropharynx, paranasal sinuses, larynx and salivary glands [3], arising mostly in males, with specific variations in the male-to-female incidence ratio from 2:1 for salivary gland tumors to 7:1 for laryngeal cancer [4]. The majority of cases are diagnosed in 50–70-year-olds; morphological diagnoses include squamous cell carcinoma (~90%) and rare cases of adenocarcinoma, melanoma and sarcoma [5,6]. The main risk factors for HNCs are smoking/drinking habits and chronic infections with HPV (human papilloma virus) or human immunodeficient virus [6].

Despite the advanced diagnostics, cancer screening and HPV vaccination programs, 60–70% of the cases are diagnosed as late as in stage III–IV associated with low life expectancy and high risks of recurrence [6]. The therapeutic options for HNC include surgical treatment, radiation therapy (RT) and chemotherapy (CT). The management tactics should be decided on personalized basis by an interdisciplinary board comprising an oncological surgeon, medical and radiation oncologists, a psychologist, a rehabilitation specialist and a dentist [3,7].

Combined-modality treatments for HNC (induction CT + surgery, surgery + postoperative RT, surgery + prophylactic irradiation of lymph nodes, chemoradiotherapy) afford 5-year overall survival rates of 50–60% [8,9,10]. However, 40–70% of the patients develop recurrence within 2 years after the treatment [11,12]. Up to 70% of the recurrent tumor volumes are locoregional, identified in the same region with the primary tumor or at its margin, i.e., within the former field of surgery or irradiation. Histories of CT and RT contribute to therapy resistance of the recurrences [13].

The main risk factors for HNC recurrence are large size of primary tumor and lymph node metastasis [14,15,16]; for RT-treated cases, these are complemented by treatment interruptions [17]. The recent advances in irradiation techniques and drug therapies notwithstanding, the latter factor remains highly relevant. Gevorkov et al. (2020) demonstrated a 27% decrease in 5-year recurrence-free survival among patients with RT interrupted for >1 month as compared to non-interrupted irradiation regimens (*p* < 0.05) [18]. According to Mollnar et al. (2021), a change in CT regimen has a negligible effect on the objective response rates in HNC (*p* = 0.47), whereas breaks in RT reduce this index from 88% to 55% (*p* = 0.003) with dramatic impact on 5-year recurrence-free survival (from 50% to 20%, *p* ≤ 0.001) [19]. In 20–30% of head and neck tumors, the treatment fails due to distant metastasis [9,20]. RT may also induce second tumors within irradiation field. A Spanish study (2020) analyzing data from 4458 pts claimed a 3.5% annual growth in the risks of second tumor. Moreover, patients with a second cancer have increased risks of further malignant neoplasms [21].

The management of recurrent tumor is challenging, with diverse treatment options and multidisciplinary teams involved. The repeated radiation therapy (reirradiation, reRT) has long entered the guidelines for recurrent HNC (rHNC). This review aimed to provide a comprehensive and updated summary of clinical outcomes of reRT for rHNC as achieved to date. An overview of relevant studies is provided in Table 1 and Table 2.

## 2. Materials and Methods

The search was conducted using PubMed, Scopus, Cochrane library and ClinicalTrials.gov to identify articles or ongoing/completed trials assessing the efficacy of reirradiation in patients with rHNC. The studies were identified using the following medical subject headings (MeSH) and keywords including: ‘‘reirradiation” and ‘‘head and neck cancer.” The search was restricted to full-text publications or abstracts in English. The Medline search strategy was: (‘‘reirradiation” (Mesh) OR ‘‘re-irradiation” (All fields)) AND (‘‘head and neck cancer” (Mesh) OR ‘‘head and neck cancers” (All fields)). The search strategy of low specificity and no restrictions on publication date were used in order to avoid missing relevant data. Articles involving none of the treatment techniques listed in Section 4 were excluded. We further focused on prospective and retrospective trials and therapeutic interventional studies reporting the outcomes in terms of overall survival (OS), or/and local control (LC), or/and progression-free survival (PFS), or/and disease recurrence, or/and treatment toxicity. 

## 3. Management of Recurrent HNC

Recurrent HNC can be treated by surgical interventions, reirradiation, following lines of CT, target- and immunotherapy [3,22]. Surgical treatment for rHNC has long become an international standard [3,22,23]. A study by Maruo et al. (2019) compared two groups of patients with rHNC after chemoradiotherapy. The first group (*n* = 75) with tumor masses in oropharynx, laryngopharynx or larynx received surgical treatment. The second group (*n* = 63) received palliative CT due to complex anatomical localization of the tumor or other contraindications for the surgery. Three-year survival rates for the two groups differed dramatically: 60.0% vs. 8.2% (*p* < 0.001) independently of the recurrent tumor localization [24]. A non-randomized study by Elbers et al. (2019) enrolled 189 patients receiving surgical treatment for post-RT recurrence of laryngeal, laryngopharyngeal or oropharyngeal cancer (in 99, 26 and 64 of the cases, respectively). Five-year local control, OS and PFS rates for the cohort constituted, respectively, 57%, 33% and 32%. A dedicated analysis revealed particular benefits of lymph node dissection affording 5-year LC rates of 72% [25].

Importantly, surgery provides the only means for the instant removal of resistant bulk tumor substantively enhancing the efficacy of adjuvant therapies. However, surgical approaches have limited applicability in rHNC: about two-thirds of the patients are inoperable due to hindered access (tumors of posterior wall of oropharynx, nasopharynx or skull base), tumor size or comorbidities with high risks of postoperative complications [26]. Many patients refuse the repeated resection as terrifying and disabling, and the usual option to resort to in such cases is CT [27]. A 2017 survey by Karabajakian et al. demonstrated the non-efficacy of different schemes of CT (single-drug or combination) in such patients: the median survival under treatment is 5–7 months, with high toxicity in 20% of the recipients [28].

A targeted option with EGFR (epidermal-growth factor receptor) inhibitor cetuximab combined to standard CT prolonged the median survival to 10–12 months without aggravation of toxicity profiles [29]. Still, the upgrade was subtle and hardly of translational significance on its own.

Another extensively studied option for rHNC involves the immune checkpoint inhibitors nivolumab and pembrolizumab. A multicenter prospective study KEYNOTE-048 (phase III, *n* = 882) compared pembrolizumab (single-drug or +CT) with cetuximab + CT regimens. Pembrolizumab monotherapy significantly improved the median survival: 14.9 vs. 10.7 months on cetuximab + CT (*p* = 0.0007). A combination of pembrolizumab with platinum drugs and 5-fluorouracil afforded a similar advantage over cetuximab + CT: 13.0 vs. 10.7 months (*p* = 0.0034), with similar toxicity rates of 85% and 83%, respectively; of note, pembrolizumab monotherapy had lower toxicity rates of 55%. Based on these data, pembrolizumab was included in international guidelines for PD-L1+ rHNC [30].

The next round of analysis (2022) accounted for PD-L1 status: <1% (subcohort 1) or 1–19% (subcohort 2). Median survival on pembrolizumab single-drug vs. cetuximab + CT constituted, respectively, 7.9 vs. 11.3 months for subcohort 1 and 10.8 vs. 10.1 months for subcohort 2. On pembrolizumab + CT vs. cetuximab + CT, median survival constituted 11.3 vs. 10.7 months for subcohort 1 and 12.7 vs. 9.9 months for subcohort 2. These data identify PD-L1 status defined as ≥1% positivity as a predictor of pembrolizumab efficacy in treatment schemes for rHNC [31].

CheckMate 141, a phase III multicenter clinical trial published in 2019, used another immune checkpoint inhibitor, nivolumab, in patients with metastatic HNC progression after previous therapy. The participants were randomized into two groups at a 2:1 ratio: the first group received nivolumab 3 mg/kg every 2 weeks and the second group received a monotherapy of methotrexate, docetaxel or cetuximab at the discretion of treating physician. The analysis was carried out with regard to the history of previous treatment with cetuximab in each particular case. According to the results, nivolumab significantly increased 1-year OS independently of the cetuximab history. In the ‘former cetuximab recipient’ subcohort, 1-year OS was 31.3% (23.9–38.9%) on nivolumab and 25.4% (16.0–35.8%) for other therapies. In the ‘former cetuximab non-recipient’ subcohort, nivolumab afforded a more substantive advantage: 1-year OS 38.5% (28.6–48.3%) vs. 11% (4.0–21.9%) on other therapies. Of note, nivolumab outperformed CT even in PD-L1 negative cases [32].

Despite the noticeable advances of CT, the outcomes are non-satisfactory and further research on new formulations and combined modalities for rHNC is needed. The surgical options for rHNC are efficacious but contraindicated or refused in/by about 60–70% of the patients for the reasons of non-resectability or severe disabling consequences.

## 4. Therapeutic Options for Head and Neck Tumor Reirradiation

Given the limitations of surgery and the limited success of drug therapies for rHNC, reRT options merit close attention. According to the British Association for Cancer Research, up to 85% of the patients with HNC receive RT as a radical or palliative treatment for the primary disease [33]. The primary RT doses are high: 46–54 Gy for prophylactic lymph node irradiation or ≥66 Gy on tumor bed and affected lymph collectors [3]. High doses of primary RT, along with anatomically intricate locations of tumors (close to critical structures), large volumes of already irradiated and changed tissues, interfere with reRT feasibility which relies on high competence of the medical team responsible for the treatment. Tumor cells that survive primary RT contribute to the pool of radio- (and probably chemo-) resistant cell clones [13,34], which undermines the reRT efficacy and appeals for non-conventional strategies [35].

In the meantime, RT is a universal therapeutic approach and the majority of irradiation techniques (brachytherapy, stereotactic body radiotherapy, 3D conformal radiotherapy including volumetric intensity-modulated options with or without rotational delivery, hadron therapies) are applicable for rHNC [3,22]. This study mainly reviewed and summarized the progress of reirradiation for recurrent HNC in recent years, in order to provide potential treatment regimens for the management of reirradiation. The subsequent paragraphs describe advantages and disadvantages for each of these techniques in rHNC, providing a summary of recommended normal tissue doses, toxicity risk factors and target volume design.

### 4.1. Brachytherapy 

Brachytherapy (BT) provides the means for local irradiation by applying radioactive isotopes to/into the foci. Positioning the sources directly in the tumor tissue and the short particle path lengths allow using high doses while avoiding pronounced irradiation of surrounding healthy tissues during the treatment.

The largest clinical study on BT in rHNC (*n* = 220) was published in 2001 by a team from several North American cancer centers. The patients were implanted with ^192^Ir seeds. The interval between primary RT and BT was 2–14 years (median 2.3 years). The predominant target sites for the treatment were neck (*n* = 118, 54%) and base of the tongue (*n* = 45, 20%). The mean irradiation volume was 68.75 cm^3^. Two- and five-year OS rates for the cohort constituted, respectively, 43% and 20%, with corresponding 2-, 5- and 10-year LC rates of 69%, 51% and 41%. Still, about one-third of the cohort (60 pts) developed severe late complications of RT, mostly soft tissue necroses (28 pts) and osteoradionecroses (17 pts). Four patients (1.8%) developed carotid rupture, which was lethal in 1 case. Of note, 65% of the secondary RT complication cases received primary RT in high total doses (>65 Gy focally) [36].

The high efficacy of BT with acceptable risks of fatal complications suggested its use in comprehensive protocols. Breen et al. (2018) reported a cohort of 69 patients receiving low-dose BT with ^125^I and ^103^Pd for rHNC. All participants had a history of external RT (45–76 Gy, median 63.6 Gy). The median interval between primary RT and BT was 1.8 years. The recurrence was localized in buccal mucosa (41 pts, 59%), cervical lymph nodes (24 pts, 35%) or in buccal mucosa spreading to cervical lymph nodes (4 pts, 6%). Fifty-eight patients of the cohort (84%) underwent tumor resection simultaneously with BT. Neck dissection was performed in 44 cases including 27/28 pts (96%) with affected cervical lymph nodes and 17/41 pts (41%) with buccal recurrence. BT with ^125^I was administered in 43 pts (62% of the cohort), the remaining 26 pts (38%) received similar treatment with ^103^Pd. Reirradiation doses varied (30–180 Gy, median 90 Gy). OS and LC rates in 1-, 3- and 5-year follow-ups constituted, respectively, 58%, 19%, 12% and 55%, 38%, 28%. Despite the supposedly local influence of BT with minimal involvement of surrounding healthy tissues, the incidence of acute and late severe complications (grade 3+) was high. Eighteen patients (27%) developed acute reactions in the form of dysphagia and xerostomia grade 3. Nineteen cases (29%) of late toxicity included grade 3 complications in 12 pts (dysphagia in 7 pts, fibrosis in 2 pts and osteoradionecrosis in 2 pts) and single cases of trismus and fibrosis grade 4. Five patients (7%) developed carotid rupture, which was lethal in 2 cases, associated with the location of radiation sources in close proximity to the vessels. Median time since the end of BT before arterial rupture was 118 days. Overall, the combination of surgery and BT improved neither LC rates nor survival, but increased the incidence of radiation complications [37].

BT was also tested as a boost for a prior course of external RT. Bhalavat et al. (2018) published outcomes for 25 patients with inoperable tumors in oral cavity (10 pts, 40%) and oropharynx (15 pts, 60%). The median irradiated volume was 85 cm^3^; 18 pts received ^192^Ir BT to a total dose (TD) of 40.5 Gy and 7 pts received it as a 27 Gy boost scheduled 2–3 weeks since the completion of external RT to 50 Gy. At median follow-up of 25 months, 1- and 2-year OS rates for the cohort constituted, respectively, 77% and 68%. Although OS rates in the two-component treatment (‘boost’) group were lower, 2-year LC rates for this group were better than with the single-step protocol (85% vs. 62%, *p* < 0.02). Toxicity reactions were reduced to altered taste perception grade 3 in 1 case and no grade 3+ complications were encountered in other patients. A more dedicated analysis of treatment parameters associated larger irradiation volume (>85 cm^3^) with reduced survival length (median 12 months vs. 26 months for the rest of the cohort, *p* = 0.02). Of note, shifting the time lapse between the courses across a threshold of 15 months increased median LC from 10 to 31 months [38]. Although BT seems to be attractive, it is considered strongly limited with tumor volume, also being technically and anatomically challenging. 

### 4.2. Stereotactic Body Radiotherapy (SBRT)

SBRT is a smart technical concept aimed to neutralize the impact of complex tumor shapes and localizations on the treatment success. A robotics adjusting for tumor geometry applies radiation in few high single doses (≥5 Gy) to a limited (usually less than 50 cm^3^) tumor volume. Such hypofractionated regimens concentrate high biologically effective doses inside the tumor and effectively damage its microcirculation [39]. The prompt delivery of the total dose (TD) allows quick return to systemic therapy crucial in patients with metastatic or multiple primary tumors [40,41].

One study (2011) enrolled 96 patients receiving reRT for rHNC by CyberKnife^®^; the follow-up lasted 2–39 months (median 14 months). The cohort was subgrouped by total dose of reRT: 15–28 Gy (29 pts), 30–36 Gy (22 pts), 40 Gy (18 pts) and 44–50 Gy (27 pts). The doses were delivered in 1–5 fractions. High total doses (>40 Gy) proved beneficial as measured by 1-|2-|3-year LC rates: 69.4%|57.8%|51.1% vs. 51.9%|31.7%|15.9%, respectively (*p* = 0.02). Target volumes < 25 cm^3^ were associated with higher chances of full response to the treatment. Late radiation toxicity reactions (dysphagia, radiation fibrosis syndrome) developed in three cases only (3.1%). The low incidence of radiation complications has been attributed to skip-a-day fractionation mode supposedly favorable for the healthy tissue recovery [42].

Ling et al. (2016) analyzed the incidence of acute and late SBRT complications with regard to localization of the recurrence. Late complications were assessed for a selection of 227 cases with >3 months follow-up. The late complication incidence was moderate (43 pts, 18.9%) and the risks have been associated with laryngeal or laryngopharyngeal localization of the recurrence (*p* < 0.05) [43].

Apart from the positive dynamics achieved with SBRT [44], several reports describe less favorable outcomes in other cohorts. As early as in 2011, Cengiz et al. reported 8 cases of carotid blowout, 7 of them lethal, amounting to 17.8% of the cohort (46 patients reirradiated for unresectable head and neck tumors, mostly nasopharyngeal, 3–206 cm^3^ in size, median 45 cm^3^). The interval between RT courses for primary manifestation and recurrence was 3.8–306 months (median 38 months). Total dose of reirradiation constituted 18–35 Gy (median 30 Gy). Dedicated analysis revealed no significant correlations of carotid rupture with tumor size (*p* = 0.682) or interval between the treatments (*p* = 0.113). Stronger factors of the blowout included juxtaposition of the tumor and the vessel at an angle ≥ 180° (*p* = 0.073) and localization of the vessel within 100% isodose area [45].

Causes and probabilities of the carotid blowout syndrome were studied by a Japanese multicenter research team (published in 2013). A cohort of 381 patients with SBRT histories presented with carotid blowout in 32 cases (8.4%), 22 (5.8%) of which were lethal. Median time since completion of the therapy till the blowout was 5 months (total range 0–69 months). Multifactorial analysis identified two risk factors for the event: skin invasion (*p* = 0.039) and infectious/necrotic lesions within irradiated area after finishing the treatment (*p* = 0.09) [46]. The analysis enrolled a group of 72 patients with laryngopharyngeal recurrences close or juxtaposed to carotid artery. The group encompassed 12 cases of carotid rupture, all of them involving intimate juxtaposition between the tumor and the artery at an angle of ≥180°. The ulceration within irradiated area was identified as a prognostic factor for carotid blowout in this group as well (*p* < 0.001). In addition, median planned treatment volume (PTV) in patients with this crucial complication was significantly bigger than in the rest of the group (52 cm^3^ vs. 13.5 cm^3^, *p* = 0.02) [47].

SBRT was also studied as a combined-modality component. Vargo et al. published a report (2015) on 48 patients after stereotactic body reRT in combination with cetuximab target therapy. All participants had histories of primary RT for HNC (52.5–118.2 Gy, median 70 Gy) and all of them had unresectable recurrent tumors (median volume 36.5 сm^3^) at the time of commencement. The interval between RT courses was 3–423 months (median 18 months). Reirradiation schedules accounted for tumor volume with a 25 cm^3^ threshold: smaller tumors (18 pts, 38%) received single focal doses of 8 Gy in 5 fractions to TD of 40 Gy; for tumors ≥ 25 cm^3^ (30 pts), single doses were increased to 8.8 Gy and delivered also in 5 fractions to TD of 44 Gy. Cetuximab was administered in 3 doses: a week before SBRT (400 mg/m^2^), on SBRT day 1 (250 mg/m^2^) and a week after (250 mg/m^2^). At median survival of 10 months, 1-year OS|LC rates constituted 40%|37%. Multifactorial analysis revealed significantly higher OS rates in patients with low-volume recurrence (<25 cm^3^; 70% vs. 22% for the other group, *p* < 0.001). Acute radiation reactions grade 3 developed in 3 pts (6%); these included single cases of mucositis, dysphagia and radioepidermitis. Late severe radiation reactions had similar incidence: dysphagia grade 3 in 1 pt and fistulas grade 3 in 2 pts showing no association with recurrence volume. The complications were resolved by non-surgical methods. The carotid blowout syndrome developed in 1 pt only; the blowout was successfully prevented by embolization [48].

Clinical data on stereotactic body reRT are summarized in meta-analysis by Lee et al. (2020), encompassing a total of 575 cases from 10 multicenter studies. Total dose amounted to 24–44 Gy (median 30 Gy) delivered in 3–6 fractions (median 5). Irradiated volume varied as 19–103 cm^3^ (median 28.7 сm^3^). The 2-year OS and LC rates constituted, respectively, 30.0% (24.5–36.1%) and 47.3% (3.1–62.1%). The incidence of severe radiation complications (grade 3+) was 9.6% (5.0–17.6%) including grade V toxicity reactions (2.4–8.6%, median 4.6%). Of note, 5 studies reported no treatment complications [49].

### 4.3. 3D Conformal Radiotherapy and Its Modifications 

Unlike surgery, BT or SBRT, 3D conformal techniques are not limited in volume by anatomical localization and configuration of the recurrence. Particular methods include intensity-modulated therapy (IMRT) and volumetric intensity-modulated arc therapy (VMAT).

The original 3D conformal option has not been properly tested for reRT, which is a rather new clinical paradigm [50,51]. As the intensity-modulated modifications afford better distribution of the dose within irradiated volume, the original version is preserved mostly for comparative purposes.

A pioneering study by Kharofa et al. (2012) enrolled 38 patients with recurrent head and neck carcinoma, all of them with histories of RT (median 68 Gy) for primary manifestation of the disease. The interval between the two courses constituted 3–228 months (median 28). Reirradiation was carried out to a median TD of 60 Gy in combination with CT: 9 pts (24%) received 3D conformal reRT and 25 pts (76%) received IMRT. Severe late toxicity developed in 6 cases (16% of the total), with most of them falling into 3D conformal subgroup and constituting almost half of it; in IMRT group the incidence of late radiation complications was significantly lower (2 pts, 8%; *p* < 0.05). Median survival time for the cohort was 16 months and 1|3|5-year OS rates were 54|31|20% and similar between the groups [52]. High radiation toxicity indicators were also reported elsewhere [53,54,55].

IMRT and VMAT outperform 3D conformal RT in terms of tolerance by virtue of advanced dose distribution parameters [56,57]. This advantage stipulates clinical prospects of IMRT and VMAT, especially in combination with surgery and CT.

Alternative fractionation schemes were analyzed in two randomized protocols, RTOG 9911 [58] and RTOG 9610 [59], both of them using 1.5 Gy + 1.5 Gy hyperfractionation to 60 Gy TD combined to various CT. Good locoregional control was achieved in one-third of participants, with 2-year OS rates of 10% and 26%, respectively. At that, the incidence of late toxicity grade 3–4 reached 40%; the reactions included severe mucosites and osteoradionecroses. Hematological toxicity rates reached 45% and 10% of the patients developed lethal RT complications. For other cohorts, the outcomes were generally similar [26,60,61].

IMRT technology involves the integrated boost principle assumed to increase the treatment efficacy by inhibiting the tumor repopulation impact. Sokurenko et al. (2017) reported 20 pts receiving IMRT for rHNC at region-specific total doses: 70 Gy for primary site, 60 Gy on high-risk cervical lymph nodes and 51 Gy on low-risk cervical lymph nodes. OS rate since reRT completion was 48% and the complication incidence was relatively low; 2 pts died of distant progression and 2 pts died of dysphagia and septic complications after reRT [62].

However, along with the ‘encouraging’ reports on IMRT tolerance, some studies indicated the opposite. Severe toxicity levels reaching above 30–40% have been associated with nasopharyngeal localization of the foci, as this area is densely packed with critical structures interfering with the treatment benefits [63,64].

Qiu et al. (2012) studied 70 patients with recurrent nasopharyngeal tumors on IMRT to median TD of 70 Gy. Of note, this report was one of the few evaluating the burden on risk organs during reRT; according to the results, the limits were exceeded in 10% of the cases only. The median interval between primary RT and reirradiation constituted 30 months. The 1|3|5-year OS, RFS and LC rates for the cohort were, respectively, 81.4|67.4|51.9%, 81.4%|65.8|47.6% and 81.4|65.8|49.3%. At that, the incidence of radiation complications grade 3–4 was high (52 pts, 74.3%). These included posterior nasopharyngeal wall ulcerations (11 pts, 15.7%), cranial nerve neuropathy (17 pts, 24.3%), trismus (12 pts, 17.1%) and hearing loss (12 pts, 17.1%). Six patients in this cohort died of profuse nasal bleeding [65].

Similar outcomes were reported by a Chinese team (2016) for a cohort of 77 pts with nasopharyngeal rHNC. The 1|2|3-year OS and LC rates (since reRT completion) constituted, respectively, 92|68|51.5% and 89.1|76.9|66.7%. However, 50 pts (64.9%) developed late toxicity reactions grade 3–4 including mucosal necrosis in 31 cases (40.3% incidence), cranial nerve neuropathy (20 cases, 26%), trismus (18 cases, 23.4%), hearing loss (4 cases, 5.2%) and temporal lobe necrosis (7 cases, 9.1%). Of 34 deaths recorded by the time of publication, 18 were associated with radiation toxicity, which corresponds to 23.3% lethality of the treatment [66].

A large multicenter study comparing IMRT and SBRT was published in 2017. The study enrolled 414 patients with inoperable rHNC randomized into IMRT and SBRT groups. IMRT afforded higher OS rates compared to SBRT specifically in patients with either >2 years interval between RT courses or <2 years without feeding tubes or tracheostomy (*p* < 0.001). Acute radiation toxicity was more typical for IMRT than SBRT: 5.1% vs. 0.5%, respectively (*p* < 0.01), and late radiation toxicity rates in the two groups were similar [67]. Among reasons for survival difference between groups, authors mentioned older age of SBRT pts, severe treatment history, along with biological advantages of protracted regimens for a wider tumor volume at higher risk of microscopic extension. 

Helical tomotherapy (HT) has an original way of the dose delivery, ‘slice by slice’, with a strip-shaped beam. Compared to linear accelerators, HT is better able to spare normal tissue while delivering a uniform dose [67]. The largest experience dedicated to reRT of HNC was presented by Choi et al., 2018. HT with the median dose of 60 Gy received 73 pts. The group was inhomogeneous: some pts (*n* = 28) also underwent prior surgery, for 52 pts, reRT was combined with CT. Median OS and LC were 33 and 23 months, respectively. Favorable outcomes were shown in pts with >2 years after prior RT, high dose of reRT (>66 Gy) and with CT addition. Acute reactions were observed only in 14% cases. Late side effects (grad 3+) occurred in 22%, with dysphagia prevailed [68]. 

Dose distributions of conformal external-beam RTs types are shown in Figure 1.

**Table 1 cancers-15-04409-t001:** Photon reirradiation for rHNC.

Study	Period	*n* pts	Follow-Up Time	Histology	Toxicity	LC	OS	DSS/PFS	RT Type
Puthawala et al. [36]	1979–1997	220	6 m	SCC	Acute: G2|60%	2y 69%	2y 43%; 5y 20%	2y 60%; 5y 33%	BT
Breen et al. [37]	2007–2016	69	36 m	65 SCC; 3 ACC	Acute: G3|27% G4|0%; Late: G 3|19% G 4|3%	1y 55%; 3y 38%; 5y 28%	1y 58%; 3y 19%; 5y 12%	N/d	BT
Bhalavat et al. [38]	2009–2016	25	25 m	SCC	Late: G3|2%	1y 84%; 2y 75%	1y 77%; 2y 68%	1y 74%; 2y 67%	BT
Nagar et al. [53]	1991–1999	33	10 m	SCC	Acute (all): G3|10% Late (all): G3|4% Acute (skin): G3|7% Late(skin): G3|48%	N/d	1y 41%; 2y 12%	N/d	photons, CT
Spencer et al. [59]	1996–1999	79	N/A	SCC	Acute: G4|17.7%; G5|7.6% Late: G3|19.4%; G4|3.0%	N/d	2y 15.2% 5y 3.1%	N/d	photons, CT
Riaz et al. [64]	1996–2011	348	32.6 m	various	G ≥ 3| 31.3%	2y 47%	2y 25%	N/d	photons
Langendijk et al. [54]	1997–2003	34	32 m	SCC	G3-4|66%	2y 27%	2y 38%; 3y 22%	N/d	photons
Duprez et al. [69]	1997–2011	60	18.5 m	SCC 48; ACC 9; Others 3	Acute: G3|35%; G4|3% Late: G3|11.7%; G4|26.7%; G5|6.7%	1y 64%; 2y 48%; 5y 32%	1y 44%; 2y 32%; 5y 22%	N/d	photons
Takiar et al. [63]	1999–2014	227	SCC 22.5 m; others 74.7 m	SCC 173; Other 33	G3|35.4%	2y 59%	2y 51%	N/d	photons
Langer et al. [58]	2000–2003	99	23.6 m	SCC	Acute: G3|49% G4|23% G5|5% Late: G3|16.9% G4|16.9% G5|3.6%	N/d	1y 50.2%; 2y 25.9%	N/d	photons, CT
Loimu et al. [14]	2000–2007	237	51 m	SCC	Late: G3|24%	2y 84%	2y 82%	2y 89%	photons
Platteaux et al. [55]	2000–2009	51	9.5 m	SCC	Acute: G3|29.4% Late: G3|35.3%	2y 32%	2y 30%	2y 28%	photons
Kharofa et al. [52]	2001–2009	38	16 m	SCC	N/d	N/d	1y 54%; 3y 31%; 5y 20%	N/d	photons, CT
Qiu et al. [65]	2003–2009	70	25 m	SCC	N/d	2y 65.8%	2y 67.4%	N/d	photons
Kong et al. [66]	2009–2014	77	25.7 m	SCC	Late: G ≥ 3| 64.9%	N/d	1y 92%; 2y 68%; 3y 51.5%	1y 78.7%; 2y 45.5%; 3y 32.3%	photons
Ling et al. [43]	2002–2013	291	9.8 m	255 SCC; 35 ACC; 31 others	Acute: G ≥ 3| 11.3% Late: G ≥ 3|18.9%	N/d	1y 41.4%; 3y 16.6%; 5y 10.8%	N/d	photons, SBRT
Rwigema et al. [42]	2003–2008	96	14 m	SCC	Acute: G3|5.2% Late: G3|3.1%	TD 40-50Gy: 1y 69.4%; 2y 57.8%; 3y 41.1%. TD 15-36 Gy: 1y 51.9%; 2y 31.7%; 3y 15.9%	all groups: 1y 58.9%; 2y 28.4%	N/d	photons, SBRT
Cengiz et al. [45]	2007–2009	46	N/d	30 SCC; 16 others	G3|4.4%	1y—83.8%	1y–47%	N/d	photons, SBRT
Vargo et al. [48]	2007–2013	IMRT 217; SBRT 197	IMRT 8.4 m; SBRT 7.1 m	IMRT SCC 205; SBRT SCC 194; others: 12 IMRT; 3 SBRT	IMRT G3|16.6%; SBRT G3|11.7%;	N/d	IMRT 2y 35.4%; SBRT 2y 16.3%	N/d	photons, IMRT vs. SBRT
Gogineni et al. [44]	2012–2015	60	6 m	45 SCC; 15 others	Late: G3|4%	1y 79%; 2y 79%	1y 59% 2y 45%	N/d	photons, SBRT

ACC—adenocarcinoma, BT—brachytherapy, CT—chemotherapy, DSS—disease-specific survival, G—grade, IMRT—include intensity-modulated therapy, LC—local control, m—months, *n*—number, N/d—no data, OS—overall survival, PFS—progression-free survival, pts—patients, SBRT—Stereotactic body radiotherapy, SCC—squamous cell carcinoma, TD—total dose, y—year.

### 4.4. Proton Therapy 

The field of proton therapy (PT), which uses the proton beam energy to treat cancer, is increasingly expanding both technically and clinically [70,71,72,73,74]. PT advantages reflect the specific character of proton beam energy distribution known as ‘Bragg peak’ [75,76]. Passing through the body substance, protons lose less energy than photons and release the remainder at the end of the path. This feature reduces radiation doses for structures located between the tumor, and those beyond the tumor as well, which is especially important when irradiating tumors located in close proximity to critical vital structures. The tumor capacity for absorbing protons in high doses have sparing effect on surrounding healthy tissues with a prospect of better treatment tolerance [77].

Based on biological parameters of charged particles and their linear transfer index, relative biological efficiency (RBE) of protons is estimated 1.1, whereas for photons, it is 1.0. Understanding of the difference is necessary for the correct conversion of the doses and treatment planning [78,79]. PT holds promise for rHNC [80] as direct dosimetric comparisons of photon vs. proton irradiation schedules have always favored the latter. 

In a study by Kandula et al. (2013), doses received by organs-at-risk (spinal cord, brainstem, larynx, oral cavity, contralateral submandibular and parotid salivary glands) for PT were significantly lower than for IMRT (*p* < 0.002–0.057). At the same time, IMRT showed better coverage, with reduced density of hotspots and higher target dose homogeneity [81]. A dosimetric comparison of 25 rHNC radiotherapy plans using IMRT vs. intensity-modulated PT (IMPT), published in 2016 as part of NCT 02242916 clinical trial, 15 out of 22 studied contoured structures (68%) showed a significant (*p* < 0.02) dose reduction with protons. The highest difference in radiation load was observed for contralateral carotid arteries and parotid salivary glands—the dose was reduced by 85–100% [82].

Dale et al. (2016) compared 25 IMRT plans in patients with oropharyngeal rHNC with a history of PT. Doses to the anterior and posterior oral cavity, middle and lower pharyngeal constrictors, esophagus, brainstem, cerebellum and other regions of the central nervous system were significantly lower with PT, which presumably reduced the dysphagia incidence and explained the low number of patients on nasogastric feeding in this cohort [83].

Several preclinical studies demonstrate that, in addition to its physical and dosimetric advantages, proton irradiation has advantageous profiles of cellular and biological response in terms of gene and protein expression by the tumor and its microenvironment compared to photons [84,85,86,87]. Mice exposed to whole body proton irradiation had significantly higher plasma levels of tumor growth factor-β (TGF-β) compared to photon irradiation. Furthermore, photon irradiation was found to support angiogenesis thereby enhancing the probability of metastasis known to have similar regulatory mechanisms [87]. Proton beams, by contrast, cause no activation of pro-angiogenic and pro-inflammatory genes and alleviate the invasive potential of tumor cells in vitro with corresponding inhibitory effect on tumor growth in mice [85]. The inhibitory effect of proton irradiation on tumor cell motility and invasiveness involves suppression of the integrin and matrix metalloproteinase functionalities.

Lupu-Plesu et al. (2017) presented a comparative analysis of proton and photon therapies on lymphangiogenesis, inflammation and proliferative responses in head and neck squamous cell carcinoma models [88]. The convincing evidence on the biological benefits of PT may have profound clinical implications [89,90]. The clinical use of PT is steadily gaining momentum with the building of new proton centers around the globe, which ensures more facile recruitment of patients into study groups [91]. Over recent years, PT has been increasingly used for reirradiation of tumor foci, notably in the brain and head and neck organs [92,93,94].

Phan et al. (2016) retrospectively analyzed histories of 60 patients with rHNC after proton beam reirradiation, with 13.6 months median follow-up and 47.1 months median interval between the courses. Some of the patients were treated surgically, the rest received CT. Median TD constituted 61.5 isoeffective Gy (isoGy) for operated and 66.0 isoGy for non-operated cases. One- and two-year LRC|OS rates constituted 80.8|72.8% and 81.3|69%, respectively. Twelve patients (20%) developed late RT reactions grade 3 including dysphagia, xerostomia and neurotoxicity; tracheostomy was performed in 2 cases and tube feeding in 6 cases; the 1|2-year incidence rates constituted 11.9|26%, respectively. The complications entailed 2 fatalities: osteoradionecrosis of the hyoid followed by bleeding 5 months post-treatment and osteoradionecrosis of the clivus with ulceration of the posterior nasopharyngeal wall. The analysis of risk factors for acute and late radiation complications identified clinical target volume > 50 cm^3^ as a significant risk factor of severe toxicity (*p* < 0.05), whereas the impact of surgery was negligible. Authors found nothing significantly associated with better survival [95].

In the same year (2016), Romesser et al. published similar PT outcomes for a larger cohort with rHNC (*n* = 92): 1-year locoregional recurrence rate of 25.1% with corresponding 21.7% incidence of severe RT complications (grade 3+, 15 pts); two patients died of carotid rupture without signs of recurrence [96].

A multicenter study by McDonald et al. (2016) enrolled 61 patients receiving PT for rHNC or second squamous cell carcinoma. Unlike abovementioned study, at 2-year OS of 32.7% and 16.5 months median survival for the cohort, the impact of surgical intervention was significant. In particular, median survival differed as 25.1 vs. 10.3 months in, respectively, operated vs. non-operated subgroup (*p* = 0.008). Severe RT complications were encountered in 24.6% of the cohort, mostly skin lesions and osteoradionecrosis (8 pts, 15.1%) and 2 pts (3.8%) developed unilateral vision loss. In 2 cases (3.8%), the complications (liquorrhea followed by meningitis and osteoradionecrosis of the clivus) entailed death 7- and 8 months post-RT, respectively [97].

Dionisi et al. (2019) analyzed PT outcomes in 17 patients with rHNC, most of them (53%) simultaneously receiving CT. Median reirradiation dose was 60 isoGy and median follow-up constituted 10 months. By 18 months of observation, OS and LC rates constituted, respectively, 54.4% and 66.6%. Severe late complications of PT were encountered in 23.5% of the cases, most commonly (17.6%) presenting with hearing loss, leading to fatal carotid blowout in 1 pt (6%) [98].

Shuja et al. (2019) conducted a rare direct comparison of IMPT and IMRT outcomes. The study enrolled 44 patients, 26 pts (59%) receiving proton beams and the remaining 18 pts (41%) receiving photons. Median reirradiation doses constituted 61.5 isoGy for IMPT, 69.6 isoGy for IMRT and 63 isoGy for the entire cohort; still, TD over two courses constituted 130 isoGy in both groups. For IMPT, acute grade 3 complication incidence was significantly lower: 31% vs. 73% for conventional photon IMRT, respectively (*p* = 0.01). This trend was evident for all types of early reactions including dysphagias (4% vs. 39%, *p* = 0.01), mucosites (8% vs. 39%, *p* = 0.001) and dermatites (12% vs. 33%, *p* = 0.03) and reproduced in late radiation sequelae including dysphagias (5% vs. 40%, *p* = 0.01) and mucosites (9% vs. 47%, *p* = 0.02). OS, LC and distant PFS rates in the two groups were similar [99]. The results were published as an abstract, with many significant points undiscussed. 

Another direct comparison of IMRT with PT, published by a French team, enrolled 55 participants (23 pts on IMRT and 22 pts on PT) reirradiated to a 66 Gy TD. The analysis revealed significant differences in 1-|2-year PFS rates: 9.1|9.1% for IMRT and 50|22.2% for the proton beam equivalent (*p* = 0.031). OS and LC rates in PT group were somewhat better, albeit beneath significance (*p*~0.17). The incidence of severe complications in both groups was high, accounting for 43.5%; these included osteoradionecroses (4 pts), radionecroses in temporal lobes of the brain (2 pts) and fatal carotid bleedings (3 pts), without significant difference mentioned by authors [100].

Proton beam reirradiation with TD equivalent to the high dose of primary RT provides efficient means of LC. A recent study (2023) using this approach enrolled 154 pts with rHNC reirradiated to a median of 70 isoGy corresponding to the standard TD for primary head and neck tumors. However, despite 66.6|71.8% 1-year OS|LC rates, the extraordinarily high TDs led to sharp increase in the incidence of severe radiation complications to 32.6%; these included 5 cases of carotid rupture [101].

A number of studies demonstrated that RBE of protons in Bragg peak may deviate from 1.1 and even exceed it [102,103,104]. This uncertainty is due to the multifactorial nature of RBE, particularly its dependence on absorbed dose, linear energy transfer (LET) distribution heterogeneity and irradiated tissue structure. It is important to consider when planning a PT; failure to do so poses high risks to critical organs and structures in the distal portion of the radiation field and can lead to severe complications affecting the brainstem, optic nerves, optic chiasm and spinal cord. Wedenberg et al. (2013) characterized biological effectiveness of protons and photons in cell lines using parameters α and β of the linear quadratic model. In cell lines with low α/β ratio, RBE and LET values were higher. Additionally, healthy tissues with α/β ratio equated to 3 are more sensitive to LET dynamics and tend to absorb lower doses with higher RBE. These results underscore the need to account for late-reacting critical structures in distal portions of proton-irradiated area, as the neglect of higher RBE for these structures poses huge risks of radiation complications [105]. Analyzing the distal portion of Bragg peak produced by a 250-MeV proton beam in a water phantom, Chen et al. (2012) showed that LET reaches its highest value beyond the absorbed dose maximum; the difference may reach 2 mm. This consideration is highly relevant to PT of tumors in close vicinity of critical organs [106].

RBE heterogeneity was also demonstrated in clinical studies comparing the incidence and localization of late radiation complications in patients receiving extreme PT schemes. Harrabi et al. (2017) analyzed the outcomes for a cohort of 430 patients, with 276 of them receiving PT and 154 receiving conventional photon RT at similar moderate doses (median TD 54 Gy). After 30 months follow-up, the incidence of radionecrosis in PT group was 3.3% (9 pts); the complication emerged at a median time of 12 months since completion of the treatment. All cases of radionecrosis were analyzed individually by matching MRI data to PT plans. The analysis revealed significant colocalization of Bragg peaks (distal portions) with the prospective foci of radionecrosis. Analysis of RT plans was not even mentioned, due to the abstract form of publication [107].

A similar increase in the incidence of late radiation reactions with the Bragg peak approaching critical structures was reported by Zhang et al. (2017). The analysis enrolled 75 patients receiving radical chemoradiotherapy for nasopharyngeal rHNC. The techniques differed (PT in 61 pts and IMRT in 14 pts) and the doses were similar, with median TD 70 Gy. The ‘real’ RBE computed using Monte Carlo method ranged within 1.12–1.25. Among late radiation complications, the incidence of temporal lobe necrosis constituted 7% for IMRT group and 14.8% for PT. Median time of necrosis development was 34 months and its 2|5-year rates were 3.6|14.4%. Identified risk factors included Asian ancestry, V_10Gy_, V_67.5Gy_, mean dose and dose per 0.5–3 cm^3^ of temporal lobe volume. These data suggest that the universal assumption of 1.1 RBE for PT may have limited clinical consistency [108]. In Figure 2, dose distributions of different types of external-beam particle therapy are shown.

### 4.5. Fast Neutron Therapy

Fast neutron therapy (FNT) with their high RBE (mostly independent of cell cycle and oxygenation) have inspired clinical expectations over decades. Eventually, the interest faded in connection with the remarkable progress in the field of ‘conventional’ photon RT. Clinical experience of FNT for rHNC is limited to few reports published in the 1980s–1990s.

In 1986, Errington and Caterall reported 15.6 neutron Gy (nGy) FNT outcomes in 23 pts. Median survival constituted 20 months, with full tumor regression in 82% of the cases and severe complications 6 pts (26%) [109].

In 1988, a Polish team published the outcomes for a cohort of 15 pts reirradiated for rHNC to a 3.3–13.2 nGy TD as either radical or palliative treatment; median survival constituted, respectively, 13.8 and 9.7 months. Remarkably, the incidence of severe complications was minimal (1 case of skin necrosis) [110].

Another paper published in 1988 analyzed a cohort of 46 pts, apparently the largest in this series. The survival was 7.5–14.4 months depending on the degree of response to the treatment. Severe complications developed in 25% of the cases and correlated with TD [111].

One of the latest studies in this series, by Micke et al. (2000), analyzed a cohort of 26 pts with rHNC receiving FNT in 1986–1994. At objective response in >50% of the cases, median survival was 7.4 months and only 5.9% of the cohort survived longer than 2 years. Acute and late toxicity were moderate, with no grade 4 events [112].

### 4.6. Neutron Capture Therapy

Neutron capture therapy (NCT) affords the release of particles with high RBE/LET immediately within the tumor. The radiopharmaceutical (usually ^10^B boronated phenylalanine, borofalan) consumed by tumor cells splits into short-range alpha particles and ^7^Li upon irradiation with epithermal neutrons. This radiochemical principle provides a unique dose gradient with high RBE. Compared to the majority of RT schemes for rHNC, severe radiation toxicity rates for NCT are relatively low [113].

One of the pioneering clinical studies by Kankaanranta et al. (2007) enrolled 12 pts. The objective response was achieved in all cases and retained by 83% of the patients for a median of 1 year. Toxicity grade 3 developed in 2 pts (17%) [114]. The analysis subsequently expanded to 30 pts revealed the incidence of new recurrences too high to justify clinical promotion despite the acceptable treatment tolerance [115].

In a Japanese prospective study (2016), the objective response was achieved in 12/17 pts with 2-year OS|LC rates of 47|28%. About half of the cohort developed acute toxicity reactions (mucositis grade 3 and 1 case of laryngeal edema and carotid rupture) and 2 pts developed late neurotoxicity [116].

A Finnish study (2019) encompassed NCT experience in 79 pts with rHNC, 75 of them with a history of RT for primary tumors. Median survival was 10 months and 2-year OS was 21%. Importantly, total doses over 18 Gy were associated with improved 2-year OS: 46% vs. 12% for <18 Gy [117].

A recent study, JapicCTI-194640 (2021), enrolled 21 pts with rHNC previously irradiated for the primary disease to a median TD of 65.5 Gy. NCT to a median dose of 44.7 isoGy produced objective response in 71% of the cases (based on 90-day monitoring). Acute toxicity reactions were ubiquitous, including alopecia, hyperamylasemia and nausea in 95|86|81% of the cases. Late toxicity reactions included non-life-threatening hyperamylasemia grade 3+, lymphopenia (24%) and single cases of intracranial infection and osteoradionecrosis [118]. 

Equipment and treatment schemes for NCT are far away from being standardized. Thus, the interpretation of reported results is quite difficult, even in understanding given doses. 

### 4.7. Carbon ion Therapy (CIT)

Carbon ions produce high-LET Bragg peaks and afford accurate dose distributions (comparable to protons) with high RBE (comparable to neutrons) [119]. As carbon-ion centers are almost as rare as NCT facilities, clinical studies in this field are sparse.

A Japanese study (2019) enrolled 48 pts receiving C-ions for rHNC in 2007–2016, mostly to a 52.8 isoGy distributed in 12 fractions. The 2-year OS|LC rates for the cohort constituted 59.6|40.5%. An over 2-year time-gap between primary RT and reirradiation was prognostically favorable. Toxicity grade 3+ was observed in 37.5% of the cases with 1 lethal outcome [120].

Vischioni et al. (2020) published promising data on 51 pts with recurrent salivary gland adenocarcinoma receiving C-ion irradiation (median biologically effective dose 72 isoGy delivered in 3–5 isoGy fractions). The actuarial 1|2-year OS was 90.2|64.0%. The low toxicity rates (17.5% and grade 3 only) were partly explained by the long time-gap since primary RT (median 6.33 years) [121].

Gao et al. (2019) published CIT outcomes in 141 pts with recurrent squamous cell HNC. High doses of C-ions (median 60 isoGy) afforded 1 year OS of 95.9%. With 14.7 months median follow-up, toxicity grade 3+ developed in 7.1% of the cases [122].

A retrospective analysis by Held et al. (2019) encompassed 229 pts treated in 2010–2017 for rHNC of various morphology—apparently the largest cohort in this series. CIT was carried out to a 51 isoGy dose (median) delivered in 3 isoGy fractions; the median cumulative dose (+ primary RT) was 132.8 isoGy. The outcomes depended on histological identity of tumors, with median survival of 33.6 months for adenocarcinomas and 13.7 months for squamous cell carcinomas. Toxicity grade 3+ developed in 14.5% of the cases, mostly neurotoxicity and single cases of osteonecrosis and carotid rupture [123]. 

These data were used as a basis for a prospective randomized C-ions vs. IMRT trial (CARE) currently under way [124]. The only finalized comparison of CIT|PT vs. SBRT|IMRT available to-date (a retrospective analysis) reveals higher efficacy of hadron therapies (1-year OS 67.9% vs. 54.1%, respectively), albeit accompanied by higher grade 3+ toxicity rates (46% vs. 24%, respectively) [82]. 

**Table 2 cancers-15-04409-t002:** Particle reirradiation for rHNC.

Study	Period	*n* pts	Follow-Up Time	Histology	Toxicity	LC	OS	DSS/PFS	RT Type
Errington et al. [109]	1971–1983	28	N/d	SCC 9; ACC 13; others 4	no necrosis—15 pts; small necrosis—7 pts; large necrosis—6 pts	1y 58%	1y 58%	N/d	FNT
Saroja et al. [111]	1976–1985	46	9.3 m	various, non-SCC	G3|25%	2 y 50%	2y 78%	2y 44%	FNT
McDonald et al. [97]	2004–2014	61	29 m	SCC 37; Other 24	G3|13.1%; G4|3.3%; G5|4.9%	2y 19.7%	2y 32.7%	N/d	PT
Beddok et al. [100]	2012–2019	55	41.3 m	SCC	N/d	2y 18.3%	2y 42.5%	N/d	PT + photons
Romesser et al. [96]	2011–2014	92	13.3 m	52 SCC; 9 ACC; 31 others	Acute: G ≥ 3| 31.4% Late: G ≥ 3|15.8%	1y 25.1%	1y 65.2%	1y 84%	PT
Phan et al. [95]	2011–2015	60	13.6 m	SCC	Acute: G3|30% Late: G3|16.7%	1y 68.4%	1y 83.8%	1y 60.1%	PT
Dionosi et al. [98] et al.	2015–2018	17	10 m	SCC	G3|23.5%	1.5y 66.6%	1.5y 54.4%	N/d	PT
Lee et al. [101]	2013–2020	242	N/d	SCC	Acute: G3|30.2%; G4|62.4% Late: G3|32.6%; G4|1.6%; G5|2%	Fx group 1y 71.8%; quad shot group 1y 61.6	Fx group 1y 66.6%; quad shot group 1y 28.5%	N/d	PT
Kankaanranta et al. [114]	2003–2008	30	N/d	29 SCC; 1 sarcoma	Acute: G3|86% Late: G3|20%	1y 95% 2y 27%	2y 30%	N/d	NCT
Wang et al. [116]	2010–2013	17	19.7 m	11 SCC; 6 others	G3|9%	2y 28%	2y 47%	N/d	NCT
Hirose et al. [118]	2016–2018	21	24.2 m	8 SCC; 13 others	Acute: G3-4|10%	N/d	SCC 2y 58%; non-SCC 2y 100%	N/d	NCT
Hayashi et al. [120]	2007–2016	48	27 m	various, non-SCC	G3|25%; G4|25%; G5|2%	2y 40.5%	2y 59.6%	2y 29.4%	CIT
Held et al. [123]	2010–2017	229	28.5 m	124 ACC; 60 SCC; 45 others	Acute: G ≥ 3|2.3% Late: G ≥ 3|8%	1y 60%; 1.5y 44.7%	26 m	N/d	CIT
Vischioni et al. [121]	2013–2020	15	22 m	7 ACC; 2 SCC; 6 others	Acute: G3–G4| 6.7%	1y 44%; 2y 35.2%	1y 92.9%; 2y 78.6%; 3y 38.2%	N/d	CIT
Gao et al. [122]	2015–2017	141	14.7 m	106 SCC; 10 ACC; 25 others	Acute: G5|0.7% Late: G ≥ 3| 10.6%	1y 84.9%	1y 95.9%	1y 95.9%	CIT

ACC—adenocarcinoma, CIT—carbon ion therapy, CT—chemotherapy, DSS—disease-specific survival, FNT—fast neutron therapy, Fx—fractionated, G—grade, LC—local control, m—months, *n*—number, N/d—no data, NCT—neutron capture therapy OS—overall survival, PFS—progression-free survival, pts—patients, SCC—squamous cell carcinoma, TD—total dose, y—year.

## 5. Reirradiation Parameters and Target Volumes

Despite the vast shared clinical experience, optimal reirradiation parameters remain a controversial clinical issue. On the one hand, to ensure efficacy, reirradiation TD should be close to that of primary RT (≥66 Gy) and advisably exceed it to overcome the radioresistance [125]. On the other hand, exceeding the primary dose is fraught with more treatment-related complications damaging the quality of life and occasionally lethal [126,127]. Setting the dose is inevitably a trade-off between efficacy and toxicity, and even the 66 Gy TD is often unattainable.

Identifying the clinical target volume is a sophisticated task involving correct delineation of tumor boundaries using tomography data, with margins sufficient for the locoregional spread inhibition. Selective irradiation of suspicious lymph nodes should be considered with caution due to the extra toxicity risks [126].

Popovtzer et al. (2008) proposed a unified principle for setting the margins based on reirradiation outcomes for a clinical cohort of 66 pts. The target volume encompassed gross tumor volume (GTV) outlined using either contrast-enhanced computed tomography or ^18^F-fluorodeoxyglucose-positron emission tomography (^18^FDG-PET) with 5 mm margins added. The doses amounted to 46–76.8 Gy (median 64 Gy). In the follow-up, 47/66 pts (71%) developed locoregional recurrences, most of them within 95% isodose area (45/47 pts, 96%). In the remaining 2 cases, the disease emerged in contralateral lymph nodes outside the irradiated volume and none of the patients developed marginal recurrence [128].

In a retrospective analysis of treatment plans for SBRT (*n* = 89; Wang et al., 2013), the target volume was initially stripped to GTV without margins, outlined using either ^18^FDG-PET +-computed tomography (45 pts, 51%) or computed tomography only (44 pts, 49%). The incidence of recurrences for the two groups constituted, respectively, 35 and 57%. The retrospectively added correction allowing 5 mm margins around GTV increased the recurrence coverage from 11.7% to 48.2% for computed tomography and from 45% to 93.6% for ^18^FDG-PET+ computed tomography. Thus, the margins substantially alleviate the risks of locoregional recurrence, albeit at a potentially high cost of toxicity [129].

A Danish team (2023) compared GTVs based on PET, MRI and objective examination data with corresponding post-operative pathomorphological evaluation in 13 pts with head and neck tumors. The values differed by 31.9, 54.5 and 27.9%, respectively, matching in 1 case only. With the addition of 5 mm margins, the number of matches increased to 11 (of 13 cases), indirectly confirming the importance of the margins [130]. The recommended target volume design is shown in Figure 3.

A multicenter study by Caudell et al. (2018) enrolled 505 pts receiving RT in 8 clinical institutions [125]. Multifactorial analysis of OS|PFS|LC rates on treatment parameters (primary treatment, repeated surgical treatment, reirradiation doses and volumes, prophylactic irradiation of suspicious lymph nodes) revealed several patterns:(1)Irradiation of suspicious lymph nodes had negligible influence on 2-year OS|LC and radiation toxicity rates independently of surgical history;(2)Two-year OS positively correlated with TD: 49.3, 34.2 and 30.4% for ≥66, 60–65.9 and <60 Gy, respectively (*p* = 0.009), with 50.9 and 67.5% rates of locoregional recurrence for ≥66 and 60–65.9 Gy, respectively (*p* = 0.082). Escalation of the dose increased severe toxicity rates (grade 3+, 14.4 vs. 5.9% for the extremes), albeit below the level of significance (*p* = 0.126);(3)Severe toxicity rates were significantly higher in patients receiving the treatment post-operatively: 22.3 vs. 11.4% in the no-surgery group (*p* = 0.006);(4)Hyperfractionation had negligible influence on the outcomes as measured by LC, OS and acute and late radiation toxicity.

## 6. Dose Constraints for Critical Structures

The ‘costs’ of RT in terms of radiation burden on critical organs has not been addressed dedicatedly except for spinal cord. Seminal works by Ang et al. (1993) and Nieder et al. (2006) demonstrated that radiation tolerance of the spinal cord (and probably of other nervous structures) is restored to 50% of its initial value in the course of 12 months since primary RT [131,132]. Exceeding this limit is a kind of ‘taboo’ in radiation oncology due to high risks of transverse myelitis. Some recent attempts to challenge this restriction have been reported. Noi et al. (2021) published the outcomes in 32 pts with excess radiation load on the spinal cord. At a median cumulative dose of 80.7 Gy (maximum 114.79 Gy) and median follow-up of 15 months, none of the patients developed myelotoxicity [133]. However, a rigorous evaluation of the interval between the courses in terms of neurotoxicity is missing; a plausible safety threshold is 2 years. On general grounds, the longer the break, the lower the risks of complications, the higher the ‘ceiling’ for TD as the major efficacy factor. 

The carotid blowout syndrome is a characteristic reRT complication with high fatality rates. In primary RT, the load on great vessels is most often neglected, which creates a huge uncertainty for reRT planning. Garg et al. (2016) identified a cumulative dose 120 Gy in 2 courses as a safety threshold for carotid rupture based on the outcomes for a small cohort with rHNC [134].

Osteo- and chondroradionecroses constitute another group of radiation complications that undermine both the life quality and survival in patients with HNC. Bots et al. identified a cumulative dose 100–120 Gy delivered in 2 courses as a safety threshold for bone and cartilage structures [135].

The risks to epithelial and muscular parallel organs (salivary glands, masticatory muscles, etc.) have not been addressed specifically; overall, these organs showed good regeneration capacity compared with the highly sensitive nervous and skeletal structures [136]. The recommended dose constraints and toxicity risk factors are summarized in Table 3.

## 7. Discussion

Over 50% of head and neck tumors are recurrent and need second treatment [139], which should optimally reproduce the initial algorithm of surgery, RT and drug therapy. However, surgery for rHNC is often unfeasible, whereas drug monotherapy is non-efficacious. In this regard, reRT represents the most relevant clinical tool for rHNC with an impressive choice of technological options.

Brachytherapy (BT), technically an irradiative surgery, shows satisfactory efficacy and good tolerance, notably the low incidence of late radiation complications. The drawbacks of BT include limited doses, limited surgical access due to anatomical hindrance, requirement of special training for the staff and high risks of intra- and postoperative complications. BT is contraindicated for extensive recurrent tumor volumes with complex topology or spread, e.g., nerve canal invasions or hard tissue infiltrates. Due to these limitations, BT has not been widely adopted for rHNC.

SBRT, a high-precision external-beam irradiation method, is a remarkable technological achievement, despite still missing a rigorous randomized comparison with other reRT protocols. Despite the promising LC rates, SBRT outcomes are seriously compromised by high incidence of lethal radiation complications that undermine the survival and require careful justification of the treatment and a cautious follow-up. Similarly with BT, SBRT has volumetric and anatomical restrictions on total dose, which limits its clinical scope. The 3D conformal RTs that involve beam intensity modulation are efficacious (as measured by OS and LC rates) and can be combined to other modalities, notably CT schemes, but the associated radiation complication incidence remains high. The toxicity reactions are facilitated by high amounts of the low-dose ‘noise’ released upon beam modulation and also by the inhibitory effect of photons on lymphocyte functionalities [140,141]. Thus, the ‘conventional’ photon RT has apparently reached its technical and physics-based limits of safety and efficacy, while remaining an enormously demanding modality.

Proton beams represent a promising reirradiation alternative with physical and dosimetric advantages. Still, clinical benefits of PT have not been established decisively, especially in terms of late radiation toxicity. Heterogeneity of the cohorts and specific biological patterns of dose distribution complicate the interpretation of available clinical evidence. The proton vs. photon choice has not been made as yet and will require new randomized trials. Of note, current biological models for RBE and LET show limited consistency with accumulating clinical data, and the discrepancy increasingly affects the treatment outcomes. Despite these limitations and shortcomings, PT has been included in guidelines as the therapy of choice for rHNC whenever available [3,22].

Fast neutron-based options have made a ‘fast’ and non-essential contribution to rHNC irradiation experience. With histories of ^60^Co units for the primary RT, the toxicity levels were overwhelming, even though many patients did not survive until late FNT complications. The advantages of photon-based options over FNT in terms of safety and efficacy, along with the almost total extinction of fast neutron facilities, have virtually excluded the future use of FNT for head and neck tumors [142].

Another option with neutrons, NCT, is fundamentally and technically complex, understudied and non-unified to a single protocol. Comparative randomization trials of NCT with other irradiation methods are currently unfeasible, especially as the sparse clinical evidence is not encouraging [143]. NCT appears compatible with IMRT, but the prospect is more on paper [144].

C-ions, by contrast, exert a favorable combination of low toxicity and high efficacy for rHNC. In theory, carbon ions have a great influence on radioresistant cancer cells, along with reduced OARs doses. Still, the technology is very expensive and rare. This situation is not going to be changed in a near future. Randomized trials of CIT are at the stage of enrollment. The dosimetric advantages of CIT have been convincingly demonstrated in silico [82], but corresponding biological models are inconsistent for translational studies for now (similarly with PT) [145].

Total reirradiation dose for rHNC should be at least 66 Gy, which is not a ubiquitously attainable threshold, especially with hypofractionation protocols. The idea of giving such high doses is based on direct dose-outcome dependance, and overcoming radioresistance in addition. The clinical target volume should be based on comprehensive PET|MRI data with 3–5 mm linear margins to allow for subclinical invasion, developed marginal geometries and motion-related errors. 

The necessitated expansion of the treatment to the healthy surroundings surely increases the risks of toxicity, which generally depend on multiple disease-and treatment-related factors including simultaneous CT [63], surgical history [69] and localization of the recurrence [136] along with irradiation parameters: target volume [146], interval between the courses [147] and most certainly doses [148]. Most of these influences are either unquantifiable or unmodifiable under clinical circumstances. In SBRT, accounting for the radiation burden on critical structures is relieved by the continuous adjustment for tumor topography; at the same time, SBRT applied on top of classical fractionation (for primary HNC) may lead to poorly interpretable patterns of severe toxicity.

The success of any anti-cancer treatment depends on multiple individual parameters and, reciprocally, the optimization of a treatment, especially for repeated intervention, may require cautious clinical stratification and wide panel of experts opinion. To the best of our knowledge, there is only one existing consensus regarding reirradiation nasopharyngeal carcinoma [149]. A patient “chosen wisely” has greater significance than the technique or particle applied. Ward et al. (2018) analyzed the outcomes of IMRT in 412 pts with rHNC stratified by clinical parameters. The best outcomes were observed in patients having an >2-year interval between RT courses after surgical resection. Worse reRT outcomes were observed in the group of non-operated patients with good somatic status and also an >2-year interval since primary RT. The worst results were observed in the group of non-operated patients with poor functional status and early recurrence [150]. Apart from operability [151], interval between the courses and somatic status, the prognosis significantly depends on patient’s age and tumor histological group and localization [152]. Tumor volume <25 cm^3^ was identified as prognostically favorable by several independent studies with different RT modalities [42,153]. 

## 8. Conclusions

The reirradiation of recurrent head and neck tumors has been studied in a numerous of clinical aspects. The empirical parameters and identified prognostic factors outline a clear set of accessible techniques and methods. There is definitely space for methodological growth, especially as patients with poor prognosis probably need more aggressive treatments. Further progress in terms of toxicity reduction and efficacy enhancement will require large randomized clinical trials and advanced solutions in dose delivery (e.g., FLASH therapy). The prospect of unified guidelines for reirradiation in rHNC is a matter of international consensus.

## Figures and Tables

**Figure 1 cancers-15-04409-f001:**
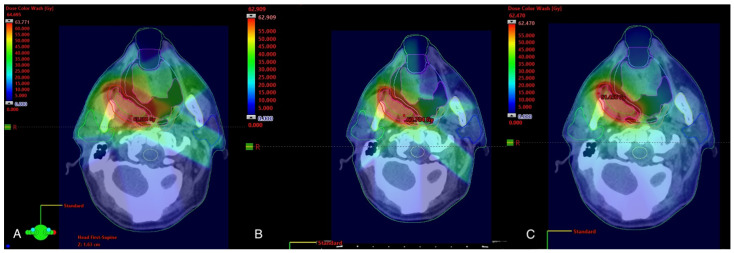
Dose distribution of 6 MeV linear accelerator for reRT of oropharyngeal relapse. (**A**) 3D-conformal treatment. (**B**) IMRT. (**C**) VMAT. Planning was calculated by Varian Eclipse ver. 16.01.04.

**Figure 2 cancers-15-04409-f002:**
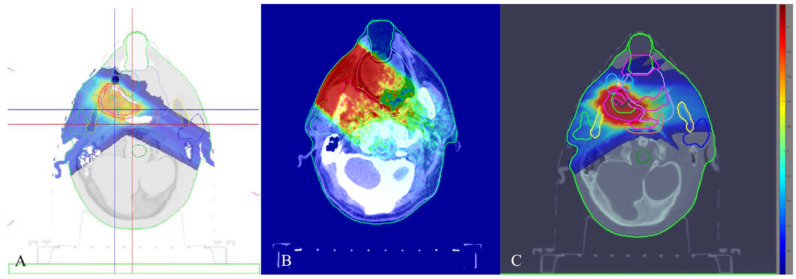
Dose distribution of 30–330 MeV proton synchrotron (**A**), 14 MeV fast neutron accelerator (**B**), 80–430 MeV carbon-ion accelerator (**C**) for reRT of oropharyngeal relapse. In silico planning was carried out with the help of ProtomPlanner ver 2.4.12 (**A**), Geant4 ver 11.1.2 (**B**) and matRad (**С**).

**Figure 3 cancers-15-04409-f003:**
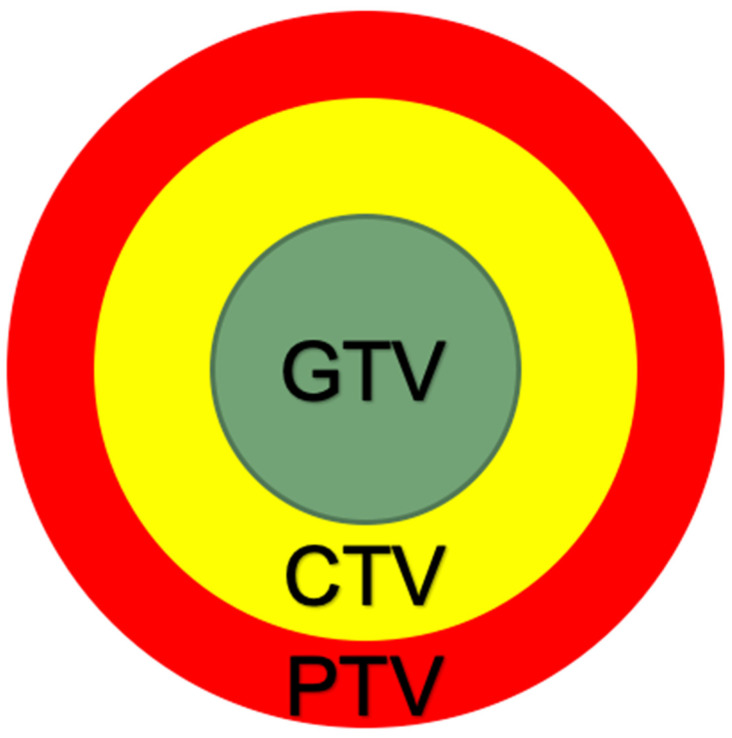
The target volume concept for reRT in rHNC, with gross tumor volume (GTV) defined by PET, MRI and objective examination; clinical target volume (CTV) allowing a margin of 5 mm; and planned treatment volume (PTV) margin depending on the immobilization efficiency (typically 3–5 mm).

**Table 3 cancers-15-04409-t003:** A summary of dose constraints for organs-at-risk and late severe side effects of reRT.

Study	Recommedation	RT Type
Ang et al. [131]; Neider et al. [132]	Nerve tissue tolerance recovered to 50% of original value 1 year since primary RT	3D conformal
Doi et al. [133]	Median cumulative dose for spinal cord—80.7 Gy; D_max_ 114.79 Gy	IMRT
Lee at al. [137]	Cumulative Dmax for brainstem 130–150% (70.2–81 Gy)	IMRT
Lee at al. [137]	Cumulative Dmax for optical structures 130–150% (70.2–81 Gy)	IMRT
Chan et al. [138]	Cumulative Dmax for temporal lobes 130–150% (91–105 Gy)	IMRT
Garg et al. [134]	Cumulative D_max_ for carotid arteria < 120 Gy	IMRT
Bots et al. [135]	Cumulative D_max_ for bone and cartilage structures < 100–120 Gy	IMRT
Yamazaki et al. [46]	Carotid blowout risk: tumor ulceration and direct contact with arterial wall > 180°	SBRT
Cengiz et al. [45]	Carotid blowout risk: direct contact of the tumor with arterial wall > 180° and vessel location within 100% isodose	SBRT
Phan et al. [95]	Late side effect risks are higher with CTV >50 cm^3^	PT
Shuja et al. [99]	Late side effect risks are higher with photons (compared to protons)	IMRT vs. PT

CTV—clinical target volume, D_max_—maximal dose at point.

## Data Availability

Not applicable.

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
