# Peer review of "An Overview of Head and Neck Tumor Reirradiation: What Has Been Achieved So Far?"

_cancers, 2023, doi:10.3390/cancers15174409_

Round 1
Reviewer 1 Report
Abstract was written well and the summary is effective
Introduction was very primitive with no main aim
Materials and Methods un-clear and useless
‘’ . Treatment protocols for rHNC’’ it seems as investigation with no clear conclusion or treatment protocol
In general more figures and diagrams should be added to clarify the review idea and explain the information
Discussion is good
Conclusion is need more precise
Reviewer 2 Report
Thank you for submitting this review of Head & Neck Tumors Reirradiation.
Reirradiation (reRT) has been proven as an effective treatment with the inclusion for the recurrent head & neck tumors in the most guidelines. The topic is interesting and worthy of publication. In this paper, the authors described its new technologies and present its clinical application. Thus, this study has certain potential clinical significance. My detailed comments are as follows:
1. The paper should be checked carefully for the unsuitable spelling/format in the word.
Line 13: Insert space in “within2 years”
Line 31: In the abstract there is an unnecessary “.” symbol at the end of the sentence, please remove. Use the Abbreviation HNCs (just introduced). Take a look at the rest of the manuscript.
Line 46: Space is missing at the beginning of many paragraphs such as this one. Please check the rest of the document.
Line 65: The p should be italicized p. Please check it carefully.
Table 1: Some abbreviations or terms were not explained in the main text or in the tables, please spelled out abbreviations such as “n pts”, “LCC” and “OS”. The notations like “-” are very confusing and use the same symbol throughout. Please check the rest of the document.
2. Parts of the methodology section could be refined considerably. The section only described the process of literature retrieval. Whether all the retrieved literatures were included in the analysis, if not, what were the procedures and criteria for literature screening and the final literature included in the analysis? Whether there were original search records. Please include detailed methods.
3. As an excellent treatment, Helical tomotherapy (HT) has been used for Reirradiation of Head and Neck Tumors. However, the authors did not describe this in the manuscript.
Reviewer 3 Report
In the article "An Overview of Head Neck Tumors Reirradiation: What Has Been Achieved So Far?" Gordon et al. analyzes in detail from a historical point of view the evolution and results of different re-irradiation methods (reRT) in recurrent head and neck cancers. The article is cursive, pleasant to read and detailed. I appreciate the approach of some less known methods (fast neutron therapy and neutron capture therapy) and evaluated than the already famous "state of the art" methods proton beam therapy and carbon ions therapy. The study focuses on the dosimetric data and the definition of the volumes for reirradiation, but it points out very correctly the idea of "chosen wisley" that must govern the implementation of the method on a large scale. It also leaves open the horizon for the future ("next level" in reRT) involving the association of reRT with immunotherapy, chemotherapy and target therapies stimulating synergistic mechanisms. I believe that the article could be published in this form.
Round 2
Reviewer 1 Report
The authors succeeded in answering all the revision comments
Reviewer 2 Report
The authors have satisfactorily responded to all my questions and made the necessary changes to the manuscript. Therefore, the manuscript is recommended for publication.